# Diagnostic performance of basal cortisol level at 0900-1300h in adrenal insufficiency

**Worapaka Manosroi** [1]*, **Mattabhorn Phimphilai**[1], **Jiraporn Khorana**[2,3], **Pichitchai Atthakomol**[4]

**1** Division of Endocrinology, Department of Internal Medicine, Faculty of Medicine, Chiang Mai University Hospital, Chiang Mai, Thailand, **2** Division of Pediatric Surgery, Department of Surgery, Faculty of Medicine, Chiang Mai University Hospital, Chiang Mai, Thailand, **3** Clinical Epidemiology and Clinical Statistic Center, Faculty of Medicine, Chiang Mai University Hospital, Chiang Mai, Thailand, **4** Department of Orthopedics, Faculty of Medicine, Chiang Mai University Hospital, Chiang Mai, Thailand

* worapaka.m@gmail.com

**Data Availability Statement:** The data that support the findings of this study are openly available at dx.doi.org/10.17504/protocols.io.7iphkdn.

**Funding:** This study was supported by the Faculty of Medicine, Chiang Mai University. The funders

## Abstract

### Objective

An ACTH stimulation test is the standard diagnostic test for adrenal insufficiency (AI). We aimed to investigate the diagnostic performance between serum morning (0800 h) cortisol and serum basal (0900–1300 h) cortisol levels and determine the proper cut-off point to facilitate AI diagnosis to reduce the number of tests.

### Methods

A six-year retrospective study was performed in a tertiary care medical center. We identified 416 patients who had undergone either low (LDT) or high dose (HDT) ACTH stimulation outpatient tests. AI was defined as a peak serum cortisol level of <500 nmol/L at 30 or 60 minutes after LDT or HDT. The associations between AI and serum basal and morning cortisol levels were demonstrated by logistic regression model. Diagnostic performance was evaluated by ROC analysis.

### Results

Of the 416 patients, 93 (22.4%) were categorized as having AI. The adjusted area under the curve (AUC) for the basal cortisol level for the diagnosis of AI was significantly higher than that for the morning cortisol (0.82 vs 0.69, p <0.001) level. The proposed cut-off values for the basal cortisol were <85 nmol/L (specificity 99.7%) and >350 nmol/L(sensitivity 98.9%). By using these proposed cut-off points, approximately 30% of the ACTH stimulation tests could be eliminated.

### Conclusion

The serum basal cortisol level with the proposed cut-off points were considered as an alternative option for diagnosis of AI. Utilizing the serum basal cortisol level can facilitate AI diagnosis as it is convenient, is not a time-specific test and has a high diagnostic performance.

had no role in study design, data collection and analysis, decision to publish, or preparation of the manuscript.

**Competing interests:** The authors have declared that no competing interests exist.

## Introduction

Adrenal insufficiency (AI) is a lethal disease if left undiagnosed. A variety of tests have been proposed to identify inadequate levels of glucocorticoid production [1]. One of the variables most commonly used to screen for AI is the serum morning cortisol level (0800 h cortisol). The dynamic tests that have been reported to properly assess AI were high-dose (HDT) and low-dose ACTH stimulation tests (LDT) [2–5]. To perform these tests, serum cortisol is drawn at 0 minutes (basal cortisol), and then either the standard 250 μg [3] or 1–5 μg [6, 7] of ACTH is administered before 30- and 60-minute serum cortisol levels are drawn.

Serum morning cortisol tests with a proper cut-off level can be used to screen for AI. If a serum morning cortisol level falls into the intermediate category, additional dynamic testing is required. Using ACTH stimulation as a reference standard for AI, some studies have proposed lower and upper cut-off values for serum morning cortisol levels of <100–145 nmol/L (3.5–5.2 μg/dL) and >375–500 nmol/L (13.6–18.0 μg/dL), respectively [8, 9]. The wide variance in the serum morning cortisol cut-off levels is caused by different cortisol assays used in multiple groups of studies, which makes it difficult to set a definite morning cortisol cut-off level. Currently, clinical practice guideline state that there is no evidence supporting the use of basal cortisol levels to diagnose AI [3]. However, using basal levels for screening has many advantages because the timing of the test is less important. Thus, it is more convenient for patients and health care providers. In fact, one study showed evidence supporting the use of basal cortisol levels to predict AI. They proposed that if the basal cortisol was >450 nmol/L (16.3 μg/dL), AI could be ruled out, with a negative predictive value of 98.7%. If the basal cortisol level was <100 nmol/L (3.6 μg/dL), AI can be diagnosed with a positive predictive value of 93.2% [10]. Still, there is no report comparing the diagnostic performance between serum morning cortisol and basal cortisol levels for the diagnosis of AI.

The objectives of our study were twofold: a) to compare the diagnostic accuracy for AI between the serum morning cortisol level and the basal cortisol level in an outpatient setting and b) to determine the appropriate cut-off values for the serum morning cortisol and basal cortisol levels that would preclude the need for dynamic testing in the outpatients settings.

## Methods

A six-year retrospective observational cohort study was performed. Data were collected from electronic medical records for all patients referred to the Adult Endocrinology Outpatient Department Unit in a tertiary care referral hospital in the northern part of Thailand between January 2010 and December 2015. The study protocol was approved by the Faculty of Medicine, Chiang Mai University, Ethics Committee. The need for consent was waived by the Ethics Committee. The patients who had undergone either LDT or HDT ACTH stimulation tests were included. Only the first tests during this period were used. The exclusion criteria were patients with incomplete data for 0800 h serum morning cortisol and ACTH stimulation testing. Women who took oral contraceptives containing estrogen, patients with suspected congenital adrenal hyperplasia and patients who had undergone pituitary surgery within the past two months were excluded. Patients receiving glucocorticoids or other traditional medicines suspected of containing glucocorticoids were told to discontinue those substances at least 24 hours before testing. Serum cortisol levels were measured by an electrochemiluminescence immunoassay (ECLIA) using an Elecsys model 1010 (Roche Diagnostic, Laval, Quebec). The intra- and inter-assay coefficients of variation for serum cortisol were <10%. In addition, the serum albumin, total cholesterol and serum creatinine data were also collected within three months before or after ACTH stimulation testing.

## ACTH stimulation testing protocol

Those who had serum morning (0800 h) cortisol levels that fell into intermediate levels of 83–499 nmol/L (3–17.9 μg/dL) were classified as either AI and normal adrenal response (non-AI) based on either LDT or HDT. All tests were performed between 0900 h-1300 h by well-trained medical nurses. The serum cortisol level was determined at 0 (basal), 30 and 60 minutes after intravenous administration of either 1 μg or 250 μg ACTH (Synacthen®, Tetracosin®). Because of the ACTH shortage in Thailand during the period from May 2010-March 2014, only LDT (1 μg) was used during that period and HDT was employed from April 2014-December 2015. ACTH 1 μg was prepared under sterile conditions by the hospital pharmacy. Briefly, a 250 μg ampule of ACTH was diluted with normal saline and transferred to a 1 ml syringe and stored at 2–8 ˚C.

## Definitions

A serum morning cortisol level sample was defined as a serum cortisol sample drawn at 0800 h while serum basal cortisol was a sample drawn between 0900 h-1300 h and before ACTH administration (0-minute cortisol). AI was defined as a peak serum cortisol level <500 nmol/L (18 μg/dL) at 30 or 60 minutes after LDT or HDT [3]. A history of glucocorticoid use was defined as the use of any type and form of glucocorticoids for at least three weeks before the tests. A history of traditional medicine use was defined as a documented history of personal use of any herbal or traditional medicine suspected of being adulterated with glucocorticoids. Symptoms of AI were defined as any symptom of fatigue, weight loss, syncope, intractable nausea and vomiting, or orthostatic hypotension documented in the medical record.

## Statistical analysis

The data were analyzed by STATA (Stata Corp., College Station, TX, USA). The statistical significance level was set as *P*-value < 0.05 for two-tailed tests. Categorical variables are presented as counts or percentages, and numerical variables are presented as means and ranges. Categorical variables were analyzed by the Fisher exact test and numerical variables by the t-test or Mann-Whitney U test, as appropriate. The 95% confidence interval (95% CI) was provided. The associations between AI and the serum morning cortisol level, serum basal cortisol level were analyzed using a logistic regression model. The areas under the ROC curves (AUC) of the model were plotted to determine the diagnostic performance of each value. Each cut-off value was calculated as the lowest (lower cut-off) or highest (upper cut-off) values, based on the highest sensitivity or specificity. The interval for each cut-off level was more than 10% of the prior cut-off levels based on the coefficients of variation of the serum cortisol test or 25 nmol/L, as appropriate. The sensitivity, specificity, PPV, NPV, likelihood ratio of positive (LHR+), likelihood ratio of negative (LHR-) and AUC were reported for each proposed cut-off point. Missing values >5% were inferred with multiple imputation analysis.

## Results

### Baseline characteristics

This study included 416 suspected AI patients (191 males and 225 females). The baseline characteristics are depicted in Table 1. Among all patients, 22.4% (n = 93/416) had a definite diagnosis of AI. The mean age was 50.0 (16.0–94.0) years. The mean age in the AI group was significantly higher than that in the no AI group. Most of the patients had the presence of AI symptoms as the indication for testing (35.3%). Those with a history of exogenous glucocorticoids use, traditional medicine use and pituitary tumor as the indication for testing had a

**Table 1.  Baseline characteristics.**

| Characteristics | Adrenal Insufficiency (n = 93) | Normal Response (n = 323) | P-Value |
|---|---|---|---|
| Age, Mean (range) (years) | 55.6 (16.0–94.0) | 48.4 (16.0–88.0) | < 0.001 |
| Gender, N (%) | | | |
| ○ Male | 40 (43.0) | 151 (46.8) | 0.524 |
| ○ Female | 53 (57.0) | 172 (53.2) | |
| Body Weight, Mean (range) (kg) | 58.4 (28.0–95.0) | 61.5 (29.6–150.3) | 0.111 |
| Underlying Diseases, N (%) | | | |
| ○ Autoimmune Diseases | 14 (15.1) | 34 (10.5) | 0.229 |
| ○ Diabetes Mellitus | 12 (12.9) | 48 (14.9) | 0.628 |
| ○ Hypertension | 25 (26.9) | 70 (21.7) | 0.298 |
| ○ Coronary Artery Disease | 11 (11.8) | 11 (3.4) | 0.001 |
| ○ Malignancy | 2 (2.2) | 6 (1.9) | 0.856 |
| ○ Others | 69 (74.2) | 205 (63.7) | 0.059 |
| ○ No known underlying disease | 1 (1.8) | 5 (1.5) | 0.823 |
| Indication for Testing, N (%) | | | |
| ○ Exogenous Steroid Use | 44 (47.3) | 62 (19.2) | <0.001 |
| - Prednisolone | 14 (27.6) | 24 (16.6) | 0.090 |
| - Dexamethasone | 1 (2.6) | 2 (1.6) | 0.685 |
| - Topical | 0 (0.0) | 3 (0.9) | 0.351 |
| - Herb or Traditional Medicine Use | 27 (45.8) | 32 (21.5) | <0.001 |
| ○ Post-Pituitary Surgery | 11 (11.8) | 69 (21.4) | 0.040 |
| ○ Pituitary Tumor | 8 (8.6) | 6 (1.9) | 0.001 |
| ○ Other Pituitary Hormonal Deficiencies | 25 (26.9) | 103 (31.9) | 0.357 |
| ○ Symptoms of Adrenal Insufficiency | 33 (35.5) | 114 (35.3) | 0.973 |
| Baseline SBP, Mean (range) (mmHg) | 119.9 (72.0–202.0) | 121.1 (78.0–199.0) | 0.646 |
| Baseline DBP, Mean (range) (mmHg) | 70.9 (83.0–113.0) | 73.4 (50.0–115.0) | 0.125 |
| ACTH Stimulation Dose, N (%) | | | |
| ○ 1µg | 37 (39.8) | 148 (45.8) | |
| ○ 250µg | 56 (60.2) | 175 (54.2) | 0.302 |
| Serum Morning Cortisol, Mean (range) (nmol/L) | 217.6 (83.8–470.6) | 266.2 (88.0–492.9) | < 0.001 |
| Serum Basal Cortisol, Mean (range) (nmol/L) | 172.7 (1.4–406.5) | 304.2 (25.7–891.8) | < 0.001 |
| Cortisol at 30 Min, Mean (range) (nmol/L) | 326.7 (44.1–491.2) | 671.6 (280.1–1749.8) | < 0.001 |
| Cortisol at 60 Min, Mean (range) (nmol/L) | 351.4 (72.6–497.6) | 743.6 (370.9–1749.8) | < 0.001 |
| Serum Albumin, Mean (range) (g/L) | 36.2 (12.0–53.0) | 39.6 (10.0–65.2) | 0.001 |
| Serum Cholesterol, Mean (range)(mmol/L) | 4.8 (1.3–8.6) | 4.8 (1.5–14.4) | 0.883 |
| Serum Creatinine, Mean (range)(µmol/L) | 94.4 (35.4–583.4) | 80.1 (26.5–822.1) | 0.098 |

SBP: systolic blood pressure

DBP: diastolic blood pressure

Range = Minimum—Maximum

higher probability of AI than no AI. Approximately 55% of the patients had undergone HDT. There was no significant difference in the AI results between LDT and HDT.

Data for the 0800 h serum morning cortisol level, basal cortisol level, 30-minute cortisol level and 60-minute cortisol level after ACTH stimulation tests categorized by the presence of absence of AI are depicted in Fig 1. In those with a normal adrenal response higher levels of cortisol were observed for all values compared to those with AI (all p-values were <0.001).

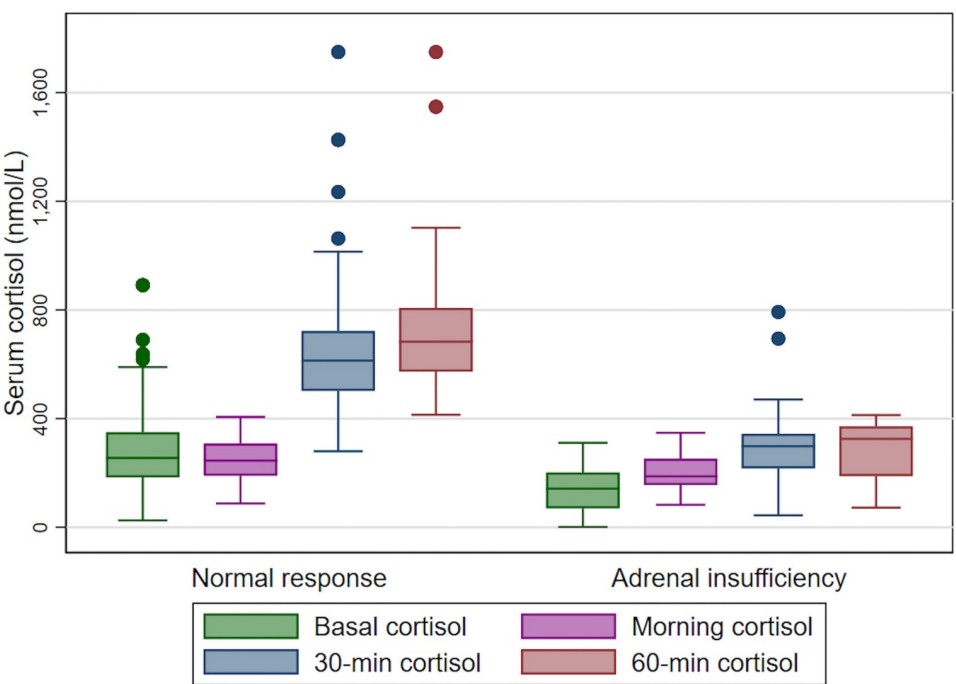

**Fig 1. Box plot graph of serum basal cortisol, serum morning cortisol, 30 and 60-minutes cortisol after ACTH stimulation test categorized by adrenal insufficiency status.**

## Diagnostic performance and cut-off values for serum morning and basal cortisol levels

The diagnostic performance of both the serum morning and basal cortisol levels after adjustment for age, sex, serum albumin level, cholesterol level, serum creatinine level and ACTH dose yielded significantly different AUCs (p <0.001). The covariate-adjusted AUC for the serum basal cortisol level was 0.82 (95% CI: 0.75–0.88), while that for the serum morning cortisol was 0.69 (95% CI: 0.62–0.78) (Fig 2). The data for univariate and multivariate analyses of the diagnostic performances of the morning and basal cortisol levels are shown in Table 2.

The proper lower cut-off value for the use of the 0800 h serum morning cortisol level to rule in AI which gave the highest specificity of 99.7% was ≤90 nmol/L (3.3 μg/dL) with a sensitivity of 4.3%. The optimal cut-off value for the serum morning cortisol level which had the highest sensitivity to rule out AI, was ≥380 nmol/L (13.8 μg/dL), with a sensitivity of 98.9% and specificity of 12.4%. The serum morning cortisol level was below the proposed lower cut-off in 1.2% (n = 5/416) of the subjects, whereas it was above the proposed upper cut-off value in 9.9% (n = 41/416). If the 0800 h serum morning cortisol cut-off value of <90 nmol/L was applied, only one patient would have received a false positive diagnosis of AI. Likewise, only one patient would have received a false negative diagnosis of AI if the 0800 h serum morning cortisol cut-off value of >380 nmol/L was applied.

For the serum basal cortisol level, the cut-off to diagnose AI with the highest specificity was ≤85 nmol/L (3.1 μg/dL), with a specificity of 99.7% and a sensitivity of 24.7% while the cut-off to rule out AI with the highest sensitivity was ≥350 nmol/L (12.7 μg/dL), with a sensitivity of 98.9% and specificity of 32.2%. Using these proposed cut-off levels, the serum basal cortisol level was below the lower cut-off point in 5.8% of patients (n = 24/416), while it was above the upper cut-off level in 25.4% (n = 106/416). If the serum basal cortisol cut-off level of <85

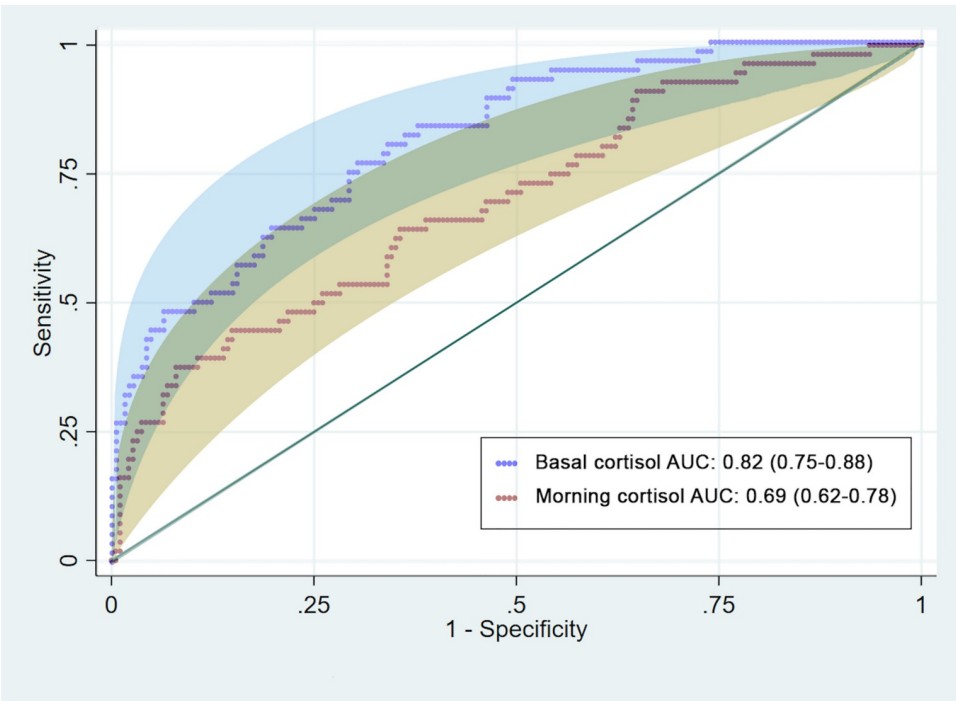

**Fig 2. Covariate-adjusted AUCs of 0800 h serum morning cortisol and basal cortisol levels for the diagnosis of AI (adjusted for age, sex, serum albumin level, cholesterol level, creatinine level and ACTH dose).**

nmol/L was used, only one patient would have been falsely diagnosed as not having AI, while if the serum basal cortisol cut-off level of >350 nmol/L was applied, one patient would have been falsely diagnosed as not having AI.

Based on the AUCs, at each cut-off level, the serum basal cortisol had higher diagnostic accuracy than the 0800 h serum morning cortisol level. The data for other cut-off levels are presented in Table 3.

## Discussion

This study highlighted a major finding, which is that the basal cortisol level between 0900 h and 1300 h has a statistically higher diagnostic performance than the serum morning cortisol level at 0800 h. Moreover, the proper cut-off values for both basal and morning cortisol levels to establish a diagnosis of AI with appropriate sensitivity and specificity were proposed.

We demonstrated that there is a diagnostic benefit of using the basal cortisol level to establish a diagnosis of AI. Additionally, the use of basal cortisol to diagnose AI has been discussed

**Table 2. Univariate and multivariate analyses for 0800h serum morning and basal cortisol levels.**

| Cortisol | Univariate analysis model | | | Multivariate analysis model* | | |
|---|---|---|---|---|---|---|
| | AUC | 95%CI | p-value | AUC | 95%CI | p-value |
| Serum morning cortisol | 0.65 | 0.58–0.71 | <0.001 | 0.69 | 0.62–0.78 | <0.001 |
| Serum basal cortisol | 0.79 | 0.74–0.85 | | 0.82 | 0.75–0.88 | |

*Multivariate analysis was adjusted for age, gender, serum albumin, cholesterol, creatinine and ACTH stimulation test doses

**Table 3. Accuracy of the cut-off level for 0800 h serum morning and basal cortisol levels.**

| | Level (nmol/L) | Sensitivity (95% CI) | Specificity (95% CI) | PPV (%) | NPV (%) | LHR+ | LHR- | TP (n) | FN (n) | FP (n) | TN (n) | AUC (95% CI) | Adjusted AUC (95%CI) |
|---|---|---|---|---|---|---|---|---|---|---|---|---|---|
| **Serum Morning Cortisol** | | | | | | | | | | | | | |
| - Lower | <90 | 4.3 (1.2–10.6) | 99.7 (98.3–100) | 80.0 | 78.3 | 13.89 | 0.96 | 4 | 89 | 1 | 322 | 0.52 (0.50–0.54) | 0.66 (0.56–0.74) |
| | <100 | 7.5 (3.1–14.9) | 99.1 (97.3–99.8) | 70.0 | 78.8 | 8.10 | 0.93 | 7 | 86 | 3 | 320 | 0.53 (0.51–0.56) | 0.66 (0.57–0.75) |
| | <125 | 15.1 (8.5–24.0) | 96.0 (92.6–97.3) | 51.9 | 79.7 | 3.74 | 0.89 | 14 | 79 | 13 | 310 | 0.56 (0.52–0.59) | 0.67 (0.58–0.76) |
| | <150 | 20.4 (12.8–30.1) | 88.5 (84.6–91.8) | 33.9 | 79.4 | 1.78 | 0.90 | 19 | 74 | 37 | 286 | 0.54 (0.50–0.59) | 0.66 (0.56–0.75) |
| | <175 | 33.3 (23.9–43.9) | 83.3 (78.8–87.2) | 36.5 | 81.3 | 1.99 | 0.88 | 31 | 62 | 54 | 269 | 0.58 (0.53–0.64) | 0.67 (0.58–0.76) |
| - Upper | >380 | 98.9 (94.2–100) | 12.4 (9.0–16.5) | 24.5 | 97.6 | 1.13 | 0.09 | 92 | 1 | 283 | 40 | 0.56 (0.54–0.58) | 0.68 (0.60–0.76) |
| | >340 | 93.5 (86.5–97.6) | 22.3 (17.9–27.2) | 25.7 | 92.3 | 1.20 | 0.29 | 87 | 6 | 251 | 72 | 0.58 (0.54–0.61) | 0.69 (0.60–0.77) |
| | >300 | 84.9 (76.0–97.5) | 34.4 (29.2–39.8) | 27.1 | 88.8 | 1.29 | 0.44 | 75 | 14 | 212 | 111 | 0.60 (0.55–0.64) | 0.71 (0.63–0.78) |
| | >270 | 68.8 (58.4–78.0) | 44.3 (38.8–49.9) | 26.2 | 83.1 | 1.23 | 0.70 | 64 | 29 | 180 | 143 | 0.57 (0.51–0.62) | 0.66 (0.57–0.75) |
| **Serum Basal Cortisol** | | | | | | | | | | | | | |
| - Lower | <85 | 24.7 (16.4–34.8) | 99.7 (98.3–100) | 95.8 | 82.1 | 79.88 | 0.76 | 23 | 70 | 1 | 322 | 0.62 (0.58–0.67) | 0.72 (0.63–0.81) |
| | <100 | 24.7 (16.4–34.8) | 99.4 (97.8–99.9) | 92.0 | 72.1 | 39.94 | 0.76 | 23 | 70 | 2 | 321 | 0.62 (0.58–0.66) | 0.72 (0.63–0.81) |
| | <125 | 31.2 (22.0–41.6) | 96.3 (93.6–98.1) | 70.7 | 82.9 | 8.39 | 0.71 | 29 | 64 | 12 | 311 | 0.64 (0.59–0.69) | 0.73 (0.64–0.82) |
| | <150 | 36.6 (26.8–41.2) | 91.6 (88.1–94.4) | 55.7 | 83.4 | 4.37 | 0.69 | 34 | 59 | 27 | 296 | 0.64 (0.59–0.69) | 0.73 (0.65–0.82) |
| | <175 | 50.5 (40.0–61.1) | 83.9 (79.4–87.7) | 47.5 | 85.5 | 3.14 | 0.59 | 47 | 46 | 52 | 271 | 0.67 (0.62–0.73) | 0.76 (0.68–0.84) |
| - Upper | >350 | 98.9 (94.2–100) | 32.2 (27.1–37.6) | 29.6 | 99.0 | 1.46 | 0.03 | 92 | 1 | 219 | 104 | 0.66 (0.63–0.68) | 0.76 (0.70–0.83) |
| | >310 | 93.5 (86.5–97.6) | 41.8 (36.4–47.4) | 31.6 | 95.7 | 1.61 | 0.15 | 87 | 6 | 188 | 135 | 0.68 (0.64–0.71) | 0.78 (0.71–0.84) |
| | >275 | 86.0 (77.3–92.3) | 50.8 (45.2–56.4) | 33.5 | 92.7 | 1.75 | 0.28 | 80 | 13 | 159 | 164 | 0.68 (0.64–0.73) | 0.76 (0.68–0.83) |
| | >250 | 80.6 (71.1–88.1) | 57.3 (51.7–62.7) | 35.2 | 91.1 | 1.89 | 0.34 | 75 | 18 | 138 | 185 | 0.69 (0.64–0.74) | 0.75 (0.68–0.83) |

PPV: Positive predictive value

NPV: Negative predictive value

LHR+: Likelihood ratio positive

LHR-: Likelihood ratio negative

TP: True positive

FN: False negative

FP: False positive

TN: True negative

in multiple studies [3, 10]. Intriguingly, our study found novel data that the basal cortisol level has a statistically higher diagnostic accuracy for AI than the morning cortisol level. In our opinion, the basal cortisol level determined from samples taken during the period 0900 h-1300 h can be used as an alternative method for screening patients for AI, obviating the need to perform serum morning cortisol testing at a specific timepoint (0800 h), resulting in more flexibility and applicability in clinical practice. However, using the AUC alone to compare the diagnostic accuracy may mislead clinicians. The results provided were only the diagnostic accuracy and the comparison of that accuracy. However, in real life, clinicians may be interested in using the lower and upper cut-off levels to rule in and rule out AI and should know the error that may occur when using these cut-off levels.

We have proposed the appropriate diagnostic cut-off values for both the morning and basal cortisol levels. The populations with suspected AI who had a serum morning cortisol levels between 83–499 nmol/L (3–17.9 μg/dL) were included. It was demonstrated that 22.4% of those who were tested with ACTH stimulation failed the test. Therefore, more than two-thirds of unnecessary procedures could be eliminated by defining the appropriate cut-off points for unstimulated serum cortisol levels. Our study established the optimal cut-off for the 0800 h morning cortisol levels as <90 nmol/L (3.3 μg/dL) and >380 nmol/L (13.8 μg/dL) for the lower and upper values, respectively. Moreover, the proper cut-off levels for basal cortisol were <85 nmol/L (3.1 μg/dL) and >350 nmol/L (12.7 μg/dL), respectively. The lower cut-off value with the highest specificity was selected to reduce the number of patients falsely diagnosed with AI because treating AI with physiologic dose glucocorticoids may harm the patients who do not need the treatment. Likewise, as AI is a lethal and life-threatening disease if left undiagnosed, the upper cut-off level to rule out AI that had the highest sensitivity was chosen to reduce the number of false negative diagnoses. Only one patient fell into the false negative groups, and one was categorized in the false positive group when the proposed upper and lower cut-off levels were employed. This demonstrated the very low risk of error. However, as having a higher sensitivity may lead to a greater number of false positive patients who may need further ACTH stimulation tests, we suggest that these proposed values may need to be applied together with clinical symptoms to form the basis for clinical decisions regarding further investigations. Our proposed upper cut-off value was different from the conventional cut-off level and from the values proposed in other studies [8–10]. Of note, the difference could be, in part, the results of the diverse population, variability in the serum cortisol assays performed and differences in the criteria for the diagnosis of AI applied in each study. If our proposed cut-off levels for serum morning and basal cortisol levels were employed instead of the conventional cut-off levels, nearly 10% and 30% (true positive and true negative rate) of the number of ACTH stimulation tests could be reduced, respectively, while maintaining high levels of sensitivity and specificity. The reduction in the number of dynamic tests needed was in concordance with the findings of other studies [9, 10]. According to our data, basal cortisol levels had a statistically higher diagnostic performance than morning cortisol levels. Hence, in terms of using the proposed cut-off levels, the basal cortisol tests appear to be an option for use in diagnosis of AI.

The present study diagnosed AI based on a peak cortisol <500 nmol/L (18 μg/dL) which was based on the first-generation cortisol assay [3]. The newer second generation assay (Cortisol II) used in other studies revealed an approximately 30% lower serum cortisol level which may lead to the overdiagnosis of AI if the conventional cut-off level was employed [11, 12]. Therefore, our proposed lower and upper cut-off levels may not be applied to the newer assays or the first-generation cortisol assays other than the Roche Diagnostic assay. Thus, we recommend that the basal cortisol levels with the proposed lower and upper cut-off levels be

employed as the screening test option and that AI symptoms should be taken into consideration before proceeding to further dynamic tests.

Our study has multiple strengths. We have demonstrated novel finding that the application of basal cortisol levels results in a high diagnostic value. Moreover, the optimal diagnostic cut-off for the basal cortisol level has been presented if this value was applied in clinical practice, it would be possible to avoid many unnecessary dynamic tests. This study also had a large number of patients, which makes the results broadly applicable to real-world practice. The confounding variables have been appropriately adjusted for in multivariable analysis, including serum creatinine, cholesterol and albumin levels, which multiple studies have demonstrated their associations with the changes in serum cortisol levels [13–15].

We acknowledge some limitations in our study. Both LDT and HDT were included in our study; thus, there was a lack of uniformity in the test procedures. In our institution, we have no specific criteria to choose whether LDT or HDT is appropriate for a specific group of patients. Although both LDT and HDT are still the standard dynamic tests that have been endorsed in many institutions [10, 16–19], whether the difference in the dosage of ACTH can influence the outcomes especially when using the same cut-off levels for both LDT and HDT to diagnose AI has not been definitively demonstrated. This might lead to the inaccuracy of the outcomes However, the ACTH dosage was used as one of the co-variables in our final model, and the potential confounding from dosage variability was thereby reduced. The gold standard test (insulin tolerance test) was not performed in our cohort. Therefore, patients with partial or recent secondary AI may have been misdiagnosed. Additionally, the population in this study was composed of patients with indeterminate results of the 0800 h serum cortisol level (83–499 nmol/L). Thus, these results may be applied only to this subgroup of the population with equivocal serum cortisol levels. Another limitation is that a diverse group of patients with both primary and secondary AI were included. Thus, the results may not be accurate for some subgroups of populations. On the other hand, this limitation makes the results more generalizable to the population usually encountered in clinical practice. Another limitation was that the exact times at which the basal cortisol level and ACTH stimulation test results were obtained were not provided, although most of the patients were tested before noon. Because the serum cortisol level follows a diurnal pattern and the timing of the test could affect the results [11], this limitation may alter the outcomes. However, a prior study stated that a single measurement of serum cortisol at 0800–1200 h can demonstrate a high sensitivity of >95%. Therefore, testing for the serum cortisol level in this proposed time range may still be beneficial in terms of convenience and flexibility [20]. The retrospective nature of our study was another limitation. Accordingly, the findings of this study should be validated in future prospective studies.

In conclusion, the basal cortisol level is considered a good alternative screening test for diagnosis of AI, with a diagnostic value which is not inferior to 0800 h serum morning cortisol level. Although, a number of ACTH stimulation tests are still required, utilizing this value makes the diagnostic procedure even easier. The proper cut-off values for the basal cortisol level were suggested, and these values have very high sensitivity and specificity. Apart from the number of dynamic tests that can be omitted, performing basal cortisol level tests is convenient, not time-dependent and can reduce expenses. Larger prospective studies in the future may be needed to confirm and validate our results.

## Acknowledgments

The authors are grateful to G. Lamar Robert, PhD and Chongchit Robert, PhD for reviewing the manuscript.

## Author Contributions

**Conceptualization:** Worapaka Manosroi.

**Data curation:** Worapaka Manosroi.

**Formal analysis:** Worapaka Manosroi, Jiraporn Khorana, Pichitchai Atthakomol.

**Methodology:** Worapaka Manosroi.

**Writing – original draft:** Worapaka Manosroi.

**Writing – review & editing:** Worapaka Manosroi, Mattabhorn Phimphilai, Jiraporn Khorana, Pichitchai Atthakomol.

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
