## [Decision Letter · Decision Letter 0]

16 Aug 2019

PONE-D-19-19551

Basal cortisol level at 0900h-1300h has higher diagnostic performance than 0800h cortisol level to diagnose adrenal insufficiency

PLOS ONE

Dear Dr. manosroi,

Thank you for submitting your manuscript to PLOS ONE. After careful consideration, we feel that it has merit but does not fully meet PLOS ONE’s publication criteria as it currently stands. Therefore, we invite you to submit a revised version of the manuscript that addresses the points raised during the review process.

The authors address a potential important practical issue regarding the timing of cortisol measurement.

The reviewers have been rather divided on this paper, but I feel it may be suitable for publication if thoroughly revised.

In addition to the points raised by the reviewers, please focus on the difference in measurement of cortisol depending on assay methods. I presume that the assay used was Elecsys® Cortisol II assay (Roche Diagnostics GmbH, Mannheim, Germany), is the suggested cut-off values for this assay correct? You may consult Vogeser, M., Kratzsch, J., Bae, Y. J., Bruegel, M., Ceglarek, U., Fiers, T., ... & Suhr, A. C. (2017). Multicenter performance evaluation of a second generation cortisol assay. Clinical Chemistry and Laboratory Medicine (CCLM), 55(6), 826-835.

Furthermore, please use SI units (with conventional units in parenthesis) and carefully proof read the manuscript before re-submission. The paper could benefit of professional English proof reading. I can recommend AJE https://www.aje.com/.

We would appreciate receiving your revised manuscript by Sep 30 2019 11:59PM. To enhance the reproducibility of your results, we recommend that if applicable you deposit your laboratory protocols in protocols.io, where a protocol can be assigned its own identifier (DOI) such that it can be cited independently in the future. For instructions see: http://journals.plos.org/plosone/s/submission-guidelines#loc-laboratory-protocols

We look forward to receiving your revised manuscript.

Kind regards,

Pal Bela Szecsi, M.D. D.M.Sci.

Academic Editor

PLOS ONE

Journal Requirements:

3. Thank you for including your ethics statement:

"The study protocol was approved by the local Research Ethics Committee".

i) Please amend your current ethics statement to include the full name of the ethics committee/institutional review board(s) that approved your specific study.

ii) Once you have amended this/these statement(s) in the Methods section of the manuscript, please add the same text to the “Ethics Statement” field of the submission form (via “Edit Submission”).

This study was supported by the Faculty of Medicine, Chiang Mai University.

Reviewers' comments:

Reviewer's Responses to Questions

**Comments to the Author**

1. Is the manuscript technically sound, and do the data support the conclusions?

Reviewer #1: No

Reviewer #2: Partly

2. Has the statistical analysis been performed appropriately and rigorously? 

Reviewer #1: No

Reviewer #2: Yes

3. Have the authors made all data underlying the findings in their manuscript fully available?

Reviewer #1: Yes

Reviewer #2: No

4. Is the manuscript presented in an intelligible fashion and written in standard English?

Reviewer #1: Yes

Reviewer #2: No

5. Review Comments to the Author

Reviewer #1: Manosroi et al. have examined the diagnostic performance of the time 0 cortisol drawn during a standard or low dose ACTH stimulation test undertaken between 0900h-1300h compared to a 0800h cortisol, to diagnose adrenal insufficiency in a patient cohort from Thailand. They conclude that a cortisol drawn between 0900h-1300h has a higher diagnostic performance than an 0800h cortisol in diagnosing adrenal insufficiency. Specific comments:

1. The main limitation of the study is the inappropriate use of a cortisol cut-off of 500 nmol/L in the ACTH stimulation test with the Roche Electrosys Cortisol II assay. Two separate publications have suggested a much lower cut off of 350 nmol/L (12.7 mcg/dL, Kline et al. Clin Biochem 2017) and 374 nmol/L (13.6 mcg/dL, Raverot et al. Ann Endocrinol 2016). Therefore using the 500 nmol/L or 18 mcg/dL as generically recommended in Ref 3 will markedly over-estimate the true prevalence of adrenal insufficiency in this cohort. It is now well established that assay-specific reference ranges are required to be used (El-Farhan et al. Clin Endocrinol 2013). This then makes the interpretation of any relationship between diagnosed adrenal insufficiency and basal cortisol (be it at 0800h or 0900-1300h) almost meaningless.

2. None of the cortisol data are depicted, the only figure is the ROC curve. Could the authors clarify if the criteria for a passed test was a cortisol of >18 mcg/dL at either 30 or 60 minutes, with the same criteria for both low dose and high dose tests?

3. Generally when establishing reference intervals, the 95% confidence interval in a normal population is used, so the upper cut point would be a 97.5% CI. Why did the authors use a >99% sensitivity/specificity when proposing their upper and lower cut points?

4. The findings are of interest, and the potential flexibility of being able to usefully interpret a cortisol drawn any time before 1300h would be beneficial for patients and clinicians alike. The main difference in the study between the "morning" cortisol and "basal" cortisol was the upper cut off of 16 vs 12.5 mcg/dL. My suspicion is that many of the patients who failed the ACTH stimulation test whose "morning" cortisol was between 12.5 mcg/L and 16 mcg/L probably in fact had normal function if an appropriate stimulated cortisol cut-off was applied.

Reviewer #2: 1) See attached file -> It is not clear why analysis was adjusted for cholesterol and creatinine.

2) Please see attached file.

3) If I am correct, there was no reason stated why the data should be available only upon request.

4) Please see attached file.

6. PLOS authors have the option to publish the peer review history of their article (what does this mean?). If published, this will include your full peer review and any attached files.

Reviewer #1: No

Reviewer #2: Yes: Tristan Struja

---

## [Author Response · Author response to Decision Letter 0]

27 Aug 2019

Responses to reviewers

We thank the reviewers for their careful reading of the manuscript and their constructive comments. We have taken all the comments to improve and clarify the manuscript. Please find below a detailed point-by-point response to all comments (reviewers’ comments in bold black, our replies in non-bold black). 

Since the reordering and restructuring of the manuscript was substantial, we have written bullet points of our major changes to the manuscript. Also, line numbering was referred to the revised manuscript without track changes.

• The manuscript has been edited by English language editing service (please see the attached certificate from AJE)

• In order to reduce the over estimation of AI prevalence, the cut-off levels to diagnose AI for serum cortisol post-ACTH stimulation test was modified to 420 nmol/L (instead of 500 nmol/L) according to proposed cut-point for second generation Cortisol II assay

• All the units have been changed to SI with conventional units in parentheses 

• Accordingly, 29 patients were excluded from the cohort, as the inclusion criteria was changed to patients with morning cortisol between 86-420 nmol/L

• Statistical power for our final sample size was still more than 80%

• The main concept and results are still the same 

• New Table 2 was created to demonstrated uni- and multivariate analyses of basal and morning cortisol

• New Table 3 for each cut-off levels was created, and multiple cut-off levels were added with the interval of 25 nmol/L or lower

• New Figure 1 was created to demonstrate and compare multiple serum cortisol values between patients with and without AI

Reviewer#1

Reviewer #1: Manosroi et al. have examined the diagnostic performance of the time 0 cortisol drawn during a standard or low dose ACTH stimulation test undertaken between 0900h-1300h compared to a 0800h cortisol, to diagnose adrenal insufficiency in a patient cohort from Thailand. They conclude that a cortisol drawn between 0900h-1300h has a higher diagnostic performance than an 0800h cortisol in diagnosing adrenal insufficiency. Specific comments:

1. The main limitation of the study is the inappropriate use of a cortisol cut-off of 500 nmol/L in the ACTH stimulation test with the Roche Electrosys Cortisol II assay. Two separate publications have suggested a much lower cut off of 350 nmol/L (12.7 mcg/dL, Kline et al. Clin Biochem 2017) and 374 nmol/L (13.6 mcg/dL, Raverot et al. Ann Endocrinol 2016). Therefore using the 500 nmol/L or 18 mcg/dL as generically recommended in Ref 3 will markedly over-estimate the true prevalence of adrenal insufficiency in this cohort. It is now well established that assay-specific reference ranges are required to be used (El-Farhan et al. Clin Endocrinol 2013). This then makes the interpretation of any relationship between diagnosed adrenal insufficiency and basal cortisol (be it at 0800h or 0900-1300h) almost meaningless.

Thank you for this very helpful comment. 

After thoroughly reviewed the literatures regarding the assay-specific cut-off levels for cortisol II assay, we totally agree with your comment (Javorsky et al. J Endocr Soc. 2019 Apr 15; Klein et al. Clin Biochem 2017; El-Farhan et al. Clin Endocrinol (Oxf).2013 May; Vogeser et al. Clin Chem Lab Med. 2017). Endocrine Society Guideline for adrenal insufficiency also stated that the appropriated cut-off for ACTH stimulation tests should be assay-specific. After discussing with all the authors, we agreed to adjust the diagnostic criteria of adrenal insufficiency by using a lower peak serum cortisol of 420 nmol/L (15.1 µg/dL) instead of 500 nmol/L which based on the data from El-Farhan et al. Clin Endocrinol (Oxf).2013. This cut-off was derived from the lower reference limit at 30 min after ACTH stimulation standardized by GC-MS method. Therefore, this could avoid the overdiagnosis prevalence of adrenal insufficiency in our population and our main results could be applied in real practice

Therefore, the analyzed data have been changed accordingly including the number of total populations. We have changed our inclusion criteria to those who had 0800hr serum cortisol between 83-420 nmol/L (3-15.1 µg/dL) instead of 83-500 nmol/L (3-18 µg/dL). Therefore, only 29 patients were excluded from the cohort. The prevalence of adrenal insufficiency was decreased from 23.8% to 21.5% and there were some changes in the proposed cut-off levels as well. However, the main idea and results are still the same. Also, the new acquired co-variate adjusted ROC for serum basal cortisol was higher than the previous result. Also, the calculated power for the new sample size was still adequate (>80%) (Bujang et al. J Clin Diagn Res. 2016 Oct). We hope that you will agree with our new peak cortisol cut-off levels for ACTH stimulation tests. 

Using 500 nmol/L for peak cortisol level is the number we normally use in our institution according to the Endocrine Society Guideline. We hope that we will change our practice of using the new cut-off levels for peak cortisol after ACTH stimulation soon. Thank you. 

2. None of the cortisol data are depicted, the only figure is the ROC curve. Could the authors clarify if the criteria for a passed test was a cortisol of >18 mcg/dL at either 30 or 60 minutes, with the same criteria for both low dose and high dose tests?

Thank you for raising this point. We have provided the data of all cortisol levels categorized by adrenal insufficiency or no adrenal insufficiency status in box-plot diagram in Fig 1(new).

The Endocrine Society Guideline of adrenal insufficiency recommended the same cut-off for peak cortisol after ACTH stimulation tests of both low and high dose tests. Therefore, the authors came to the conclusion that we will use the same criteria for adrenal insufficiency for both low and high dose ACTH test are the same which is 420 nmol/L. We have added that in the manuscript in L99-100. Also, we have used ACTH dose (low or high dose) as one of the covariates and were adjusted in multivariate model in order to reduce the variability of the dose ACTH used.

3. Generally when establishing reference intervals, the 95% confidence interval in a normal population is used, so the upper cut point would be a 97.5% CI. Why did the authors use a >99% sensitivity/specificity when proposing their upper and lower cut points?

Normally, the 95%CI was used to estimate the range of the point estimate or to document the percentile. The way we chose each cut-off level is not depend on the 95%CI of overall patients in the cohort. We manually chose a very high sensitivity (>99%) for upper cut-off level to rule-out adrenal insufficiency as this is a lethal condition if we misclassify them. Likewise, the lower cut-off to rule in adrenal insufficiency should have a very high specificity, because to prescribe physiologic dose of corticosteroid may do harm to the patients if we over-diagnose them. Therefore, we finely adjusted each cut-off value level by level with the interval of 25 nmol/L or lower, as appropriate. Therefore, each cut-off level has its own sensitivity, specificity and 95%CI.

We have revised Table3 (new) and provided the data of 95%CI for each proposed cut-off sensitivity and specificity. We also have discussed the reason why we chose a very high sensitivity and specificity for each cut-off levels in L220-230 (discussion section).

4. The findings are of interest, and the potential flexibility of being able to usefully interpret a cortisol drawn any time before 1300h would be beneficial for patients and clinicians alike. The main difference in the study between the "morning" cortisol and "basal" cortisol was the upper cut off of 16 vs 12.5 mcg/dL. My suspicion is that many of the patients who failed the ACTH stimulation test whose "morning" cortisol was between 12.5 mcg/L and 16 mcg/L probably in fact had normal function if an appropriate stimulated cortisol cut-off was applied.

Thank you for raising this issue. 

We have applied the new cut-off level for stimulated cortisol and the new proposed upper cut-off to rule-out AI in our study were >350 nmol/L (12.6 µg/dL) for 0800h morning cortisol and >337 nmol/L (12.2 µg/dL) for serum basal cortisol. Comparing to our previous results, the upper cut-off for 0800h morning cortisol has decreased from 440 to 350 nmol/L.

Further analysis

 Serum morning cortisol

Basal cortisol >350 nmol/L <350 nmol/L

>337 nmol/L 14 91

<337 nmol/L 31 306

The new upper cut-off for both morning and basal cortisol (350 and 337 nmol/L) showed smaller interval compared to our previous one (440 and 347 nmol/L)

There were 91 patients who had levels between 337-350 nmol/L. Based on our analysis, only one (1/91) patient in this group was truly diagnosed with adrenal insufficiency and the other 90 patients had normal adrenal function. This could presume that our proposed cut-off level to rule out AI particularly from basal cortisol had fairly high accuracy.

Reviewer#2

Manosroi et al. present an interesting piece of research. In my opinion, it can add substantial guidance to the community, but it needs thorough revision first.

Thank you for the positive and encouraging comments.

Minor:

1) L 50: Please provide cortisol reference ranges in international units at least the first time they appear in the text.

Thank you. We have changed all units in the text to SI with conventional units in the parentheses. 

2) L56: remove “there is” as it is unnecessary

It was removed as suggested. 

3) L73 and 74: Rewrite to ….estrogen, patients with suspected congenital adrenal hyperplasia…

Thank you. The sentence has been rewritten as suggested. 

4) L99: Rewrite “…history of personal using of…” to “…history of personal use of…”

The sentence has been rewritten as suggested.

L105 & 118 : Make sure to use the correct temporal form, multiple uses of past tense although present would be appropriate, for instance “..characteristics were depicted in Table 1.” Instead of “characteristics are depicted”.

We have corrected the sentence as suggested. Thank you.

5) Discussion: Although not a native speaker myself, I detected various semantic, and grammar mistakes in this section. They fall in the same category as those above. Please, revise your manuscript carefully.

Thank you. According to this issue, we apologize for all the grammatical mistakes and the revised manuscript has been edited by the English language editing service (please see the attached certificate from AJE).

Major: 

Table 1: Numbers of categorical variables do not add up to 471. For instance, underlying diseases adds up to 268 only. Please provide on missing data. It is ok if this piece of information is retrospectively not retrievable anymore, as long it is clearly stated.

Thank you for noticing this. 

Reviewer#1 has raised issue regarding the diagnostic cut-off point for Cortisol II electrosys assay We have thoroughly reviewed multiple literatures about this second generation assay and the specific cut-off levels for ACTH stimulation test which are lower than the usual cut-off points (Javorsky et al. J Endocr Soc. 2019 Apr 15; Klein et al. Clin Biochem 2017; El-Farhan et al. Clin Endocrinol (Oxf).2013 May). If the previous cut-off for cortisol (500 nmol/L) was used, this will markedly overestimate the real prevalence of adrenal insufficiency. Therefore, we agreed with Reviewer#1 and we have modified the diagnostic criteria for adrenal insufficiency by using the peak cortisol cut-off at 30 or 60 min at 420 nmol/L (15.2 µg/dL) instead of 500 nmol/L (El-Farhan et al. Clin Endocrinol (Oxf).2013 May). 

Therefore, the data have been modified accordingly and the new number of all population were 442 patients as only 29 patients were excluded. Some numbers have been changed but the main idea and results are still the same. Also, the power of our new sample size is still >80% (Bujang et al. J Clin Diagn Res. 2016 Oct).

We also added the data in Table 1 for patients with no known underlying disease and other diseases. Please note that the add up numbers are higher than number of total populations as some patients may have more than one underlying disease. Also, other demographic data have been revised. There was no missing value for demographic variables. For biochemical variables apart from serum cortisol (e.g. albumin, cholesterol, creatinine), the missing variables of more than 5% were dealt with multiple imputation analysis as stated in L120-121.

L193 Rewrite “Of note, the different could be” to “Of note, the difference could be”

Thank you. The sentence has been rewritten.

Table 2: Authors should provide cortisol in international units in the legend to facilitate understanding for readers from Europe. Also, it would be prudent extend the table to include various ranges of cut-offs (maybe in steps of 2ug/dl?). This way readers can judge for themselves which cut-off would be optimal for their need.

As per Plos One manuscript guideline, we have changed all units to SI units with conventional units in parentheses. 

We have modified an old Table 2 to the (new) Table 3 which included multiple cut-off values with the interval of 25 nmol/L (~1 µg/dL) or lower. 95%CI of sensitivity, specificity, adjusted AUC and non-adjusted AUC for each cut-off level were provided.

What I liked is that numbers of true negatives, true positives etc. were presented. This really helps to find out how many patients could potentially be misclassified, if the proposed cut-off were applied in practice.

Your encouraging comment is greatly appreciated. Thank you.

Authors should include a discussion how they arrived at the proposed cut-offs. Were they stated a priori or a posteriori? 

We have added the statement in the statistical analysis section L116-118. Also, in the discussion section, we have stated the reason why we have to choose the upper cut-off with the highest sensitivity and the lower cut-off with the highest specificity in L220-230. 

Furthermore, it is misleading to state an AUC of 0.82 in the abstract that is derived from figure 1, but nowhere presented (such as in table 2). Additionally, discussion should include a statement on the predictive abilities of ROC curves themselves. In case of rare events (what an AI in clinical practice gladly is), it tends to give overoptimistic results. Also, the overall AUC might not be what a reader is interested in, as it can be deceptive. Although AI is rare but potentially deadly, a clinician wants to minimize the numbers of false negatives. Therefore, it is important that these figures are presented in table 2.

We have created (new) Table 2 with data of both univariate and multivariate analyses of diagnostic accuracy for serum morning and basal cortisol. Also, both covariate-adjusted and non-adjusted AUCs were demonstrated in Table 2. 

The discussion regarding the diagnostic accuracy based on ROC alone was added on L207-211 and L223-227.

I am honestly unsure how creatinine and serum cholesterol could affect the level of cortisol. Authors should either remove these variables form analysis or clearly state why they included them. I might rather be sensible to include the dose of ACTH (LDT vs. HDT) used as a covariate.

Thank you for raising this issue. 

We agree with you on your helpful comment on the ACTH dosage as a covariate. As the ACTH dosage itself could affect serum cortisol level, therefore, we have re-analyzed the data and adjusted for ACTH dosage in multivariate analysis as suggested. 

The associations between serum cholesterol and adrenal insufficiency in cirrhotic patients have been reported in multiple literatures including human and animal studies with varying results. However, there was no report of this issue in non-cirrhotic patients (Park et al. Medicine. 2018;97(26); Spadaro et al. Scandinavian journal of gastroenterology. 2015;50(3); Ouweeneel et al. Atherosclerosis. 2017;261). As cholesterol is the precursor for glucocorticoids synthesis (Bochemet al. J Lipid Res. 2013 Jun), we assume that cholesterol levels may cause an interference with cortisol levels. We have stated this issue in the discussion part (L256-258). 

However, multiple studies had reported normal findings of adrenal function in chronic renal failure patients, physiologic changes of serum cortisol have been reported in multiple studies (Clodi et al. Am J Kidney Dis. 1998;32(1); Raff et al. Endocr Connect. 2013;2(1)). Serum half-life of cortisol is prolonged in chronic renal failure and both elevated and normal cortisol levels had been reported (Bacon et al. The Johns Hopkins medical journal. 1973; Nolan et al. JCEM. 1981;52(6); Sianmopoulous et al. Peritoneal dialysis international. 1990;10(2)). Therefore, in those with poor eGFR (high serum creatinine), serum cortisol may be altered. So, we have used serum creatinine as one of the covariates. We also have stated this issue in the discussion part (L256-258). 

Furthermore, results from unadjusted analysis are not presented. Please include them in a separate column in table 2. 

The new Table 3 for the unadjusted and adjusted data (AUCs) was created as suggested. Thank you. 

L202 I assume that Cortisol II stands for second generation cortisol immunoassays. Please elaborate more on the differences between assays.

Thank you. We also discussed more about the differences between assays and the application of the proposed cut-off levels in other assays in the discussion section. (L242-250) 

L 227 and 228 “Another limitation was the exact time at which the basal cortisol and ACTH stimulation had been tested were not provided but most of the patients were tested before noon.” This sentence questions the whole validity of the study. Please present the testing times in table 1. Table 1 should also include two additional columns splitting the population into 2 groups according to a morning cortisol group and a basal cortisol group.

Thank you for raising this issue. 

We apologized that the data of the specific time point at which basal cortisol was tested were not available in our cohort. The time was recorded in the range between 0900h and 1300h. Therefore, we put this issue as one of our limitation.

However, one study in year 2019, also demonstrated a high sensitivity (>95%) of serum cortisol testing in the time range between 0800h-1200h (Mackenzie et al. Clin Endocrinol. 2019). The discussion regarding this limitation was added in L277-281. Therefore, we presumed that to test cortisol with our proposed time range still has validity. We plan to perform a prospective study regarding multiple time testing if serum cortisol in the future. 

Each patient in our cohort had been performed both basal and morning cortisol levels as stated in the ACTH stimulation testing protocol section. Therefore, we cannot categorize the patients into 2 groups as you suggested. To performed both tests in the same patients support our results validity as this could reduce the confounders which occur from different patients. 

If I am correct, there was no reason stated why the data should be available only upon request.

After discussing with other authors, the was no restriction to our data access. We have edited that in the submission form. The URL for the data access is https://drive.google.com/open?id=1XutNq1CD57xqjOInx_VM46LLXCCkEaDa

Thank you.

---

## [Decision Letter · Decision Letter 1]

19 Sep 2019

PONE-D-19-19551R1

Basal cortisol level at 0900 h-1300 h has higher diagnostic performance than the 0800 h morning cortisol level for adrenal insufficiency

PLOS ONE

Dear Dr. manosroi,

Thank you for submitting your manuscript to PLOS ONE. After careful consideration, we feel that it has merit but does not fully meet PLOS ONE’s publication criteria as it currently stands. Therefore, we invite you to submit a revised version of the manuscript that addresses the points raised during the review process.

The manuscript has improved, but some issues remain. Please focus on the cut-off problem.

We would appreciate receiving your revised manuscript by Nov 03 2019 11:59PM. To enhance the reproducibility of your results, we recommend that if applicable you deposit your laboratory protocols in protocols.io, where a protocol can be assigned its own identifier (DOI) such that it can be cited independently in the future. For instructions see: http://journals.plos.org/plosone/s/submission-guidelines#loc-laboratory-protocols

We look forward to receiving your revised manuscript.

Kind regards,

Pal Bela Szecsi, M.D. D.M.Sci.

Academic Editor

PLOS ONE

Reviewers' comments:

Reviewer's Responses to Questions

**Comments to the Author**

1. If the authors have adequately addressed your comments raised in a previous round of review and you feel that this manuscript is now acceptable for publication, you may indicate that here to bypass the “Comments to the Author” section, enter your conflict of interest statement in the “Confidential to Editor” section, and submit your "Accept" recommendation.

Reviewer #1: (No Response)

Reviewer #2: All comments have been addressed

2. Is the manuscript technically sound, and do the data support the conclusions?

Reviewer #1: Partly

Reviewer #2: Yes

3. Has the statistical analysis been performed appropriately and rigorously? 

Reviewer #1: Yes

Reviewer #2: Yes

4. Have the authors made all data underlying the findings in their manuscript fully available?

Reviewer #1: Yes

Reviewer #2: Yes

5. Is the manuscript presented in an intelligible fashion and written in standard English?

Reviewer #1: Yes

Reviewer #2: Yes

6. Review Comments to the Author

Reviewer #1: Manosroi et al. have undertaken a substantial revision of the manuscript and it is significantly improved. Some issues remain to be addressed:

1. Ref 11 (El-Farhan et al.) did not use the Roche cortisol II assay, they used the older Roche (cortisol I) E170 assay. Their cut-off of 420 nmol/L is based on GCMS. The papers cited by the authors which did use cortisol II, namely Javorsky et al. and Kline et al. reported cortisol cut offs of 400 nmol/L and 350 nmol/L respectively (and a third study previously mentioned by this reviewer, Raverot et al. was 374 nmol/L). Therefore it is puzzling why the authors chose 420 nmol/L as their cortisol cut-off. Furthermore, since the data are from 2010-2017, it seems unlikely that the authors' laboratory used a cortisol II assay for this entire period.

2. I think it is misleading if not frankly incorrect to state that their proposed cortisol cut-offs demonstrated that the basal cortisol drawn later in the day had a higher diagnostic performance than a 0800h cortisol. Firstly neither the lower nor upper proposed cut-offs are significantly different from each other, when taking into account the cortisol assay CVs. The lower cut off of 89 vs 86 nmol/L and the upper cut-off of 350 vs 337 nmol/L are well within even a 5% CV boundary.

Secondly, even if one takes these as different numbers, on the analysis performed there was 1 false positive and 1 false negative in each group, with identical sensitivity at the upper cut-off and specificity at the lower cut-off.

Point 1 is essential to resolve in my opinion - the cut-off of 420 nmol/L is still at least 20 nmol/L higher than any published data on the cortisol II assay in the ACTH stimulation test, if this was in fact the assay used throughout the study period.

Taken on face value with the current data set, the conclusion of the study should be that a basal cortisol taken any time during the morning up to 1200h (it is unclear how many were done between 1200h and 1300h) provides useful information in assessing the HPA axis integrity, though the cut-offs are not significantly different from the 0800h cortisol.

Reviewer #2: My concerns have been addressed. I appreciate the extensive work of the authors in regard to language, statistics and readability.

7. PLOS authors have the option to publish the peer review history of their article (what does this mean?). If published, this will include your full peer review and any attached files.

Reviewer #1: No

Reviewer #2: Yes: Tristan Struja

---

## [Author Response · Author response to Decision Letter 1]

21 Sep 2019

Response to reviewers

Reviewer #1: Manosroi et al. have undertaken a substantial revision of the manuscript and it is significantly improved. Some issues remain to be addressed:

1. Ref 11 (El-Farhan et al.) did not use the Roche cortisol II assay, they used the older Roche (cortisol I) E170 assay. Their cut-off of 420 nmol/L is based on GCMS. The papers cited by the authors which did use cortisol II, namely Javorsky et al. and Kline et al. reported cortisol cut offs of 400 nmol/L and 350 nmol/L respectively (and a third study previously mentioned by this reviewer, Raverot et al. was 374 nmol/L). Therefore it is puzzling why the authors chose 420 nmol/L as their cortisol cut-off. Furthermore, since the data are from 2010-2017, it seems unlikely that the authors' laboratory used a cortisol II assay for this entire period.

Response: 

We would like to apologize for our mistake in stating the wrong generation of assay used for cortisol testings in the manuscript. After we have double checked with the former head of the Endocrinology Laboratory at our institution, the Elecsys Cortisol II assay has been used since January 2016 until present time. Therefore, the period between January 2010-December 2015, the cortisol tests were performed by electrochemiluminescence immunoassay using an Elecsys model 1010 (Roche Diagnostic, Laval, Quebec) which is an older assay. Please find the attached certify letter from the former head of Endocrinology laboratory from our institution.

As the serum cortisol in the main population was performed by first-generation assay, we decided to exclude those with second-generation assay. The overall population were 471 patients. There were 55 patients excluded and 416 patients included. Also, as the assay used was the first generation, the conventional cut-off (peak cortisol <500 nmol/L) for peak cortisol after ACTH stimulation test was used to diagnose AI and we have included those with indeterminate serum cortisol between 83-499 nmol/L (Bornstein. J Clin Endocrinol Metab. 2016 Feb). Although, the total sample size has reduced, the statistical power is still more than 80% (Bujang. J Clin Diagn Res. 2016 Oct)

We have re-analyzed all the data according to the new cut-off level to diagnose AI (peak cortisol <500 nmol/L).

- All the Tables (Table 1-3) and Figures (Figure1 and 2) have been modified accordingly. 

- In the method section, the period of data collection L72 and 75 has been changed. The method of cortisol assay used in the study has been modified (L85-86) 

- In the ACTH stimulation testing protocol section, the indeterminate cortisol levels have been adjusted to level 83-499 nmol/L (L91)

- In the definitions section, the peak serum cortisol for AI diagnosis has been changed to >500 nmol/L (L103-105)

- In the discussion section, L248-256 have been modified.

- Minimal changes of some values in baseline characterisitics, results and discussion section.

- Only minimal changes in upper and lower cut-off levels were observed with the same sensitivity and specificity. 

- The main concept of the manuscript is still the same.

We again apologized for our mistake and misunderstanding regarding the assay used in the manuscript which bring to some confusions. We hope that to include only those with older assay would benefit in terms of data homogeneity. Thank you. 

2. I think it is misleading if not frankly incorrect to state that their proposed cortisol cut-offs demonstrated that the basal cortisol drawn later in the day had a higher diagnostic performance than a 0800h cortisol. Firstly neither the lower nor upper proposed cut-offs are significantly different from each other, when taking into account the cortisol assay CVs. The lower cut off of 89 vs 86 nmol/L and the upper cut-off of 350 vs 337 nmol/L are well within even a 5% CV boundary.

Secondly, even if one takes these as different numbers, on the analysis performed there was 1 false positive and 1 false negative in each group, with identical sensitivity at the upper cut-off and specificity at the lower cut-off.

Response: 

The reason why we stated that basal cortisol had higher diagnostic performance than morning cortisol was based on the adjusted area under ROC (Fig 2). Area under ROC is the degree of agreement between the index test (basal or morning cortisol) and the reference standard (ACTH stimulation test) in which basal cortisol level showed higher rate of agreement to ACTH stimulation tests than morning cortisol (Florkowski. Clin Biochem Rev. 2008 Aug). This AuROC in Fig2 was calculated by the overall values for basal and morning cortisol, not for just the specific cut-off level. 

The upper and lower cut-off levels we have proposed for basal and morning cortisol per se did not provide data on diagnostic accuracy for the whole values. The proposed cut-off level that we have chosen to report is just the spot on the ROC curve which gave the highest sensitivity or specificity for basal or morning cortisol. Therefore, these spots cannot demonstrate the whole picture of diagnostic accuracy unlike AuROC in Fig2. (Unal. Comput Math Methods Med. 2017)

We agree with you that our proposed cut-off levels between basal and morning cortisol were almost the same number which gave the same sensitivity and specificity. However, after statistical analysis by comparing an adjusted AuROC between the lower cut-off for morning (<90 nmol/L) and basal cortisol (<85 nmol/L) as shown in the table and figure below, basal cortsol showed statistically higher AuROC (diagnostic accuracy) than morning cortisol (p<0.01). These data were not shown in the manuscript.

 AuROC p-value

Morning cortisol 0.65 0.01

Basal cortisol 0.71 

Figure demonstrated AuROC between proposed lower cut-off level for basal and morning cortisol

Likewise, the upper cut-off levels for basal (>350 nmol/L) and morning cortisol (>380 nmol/L) demonstrated significantly different in diagnostic accuracy as shown in the table below.

 AuROC p-value

Morning cortisol 0.68 0.003

Basal cortisol 0.76 

Figure demonstrated AuROC between proposed upper cut-off level for basal and morning cortisol

Also, when looking at the absolute number of patients using these cut-off levels for morning and basal cortisol, nearly 10% and 30% of the patients (true positive and true negative numbers) could bypass the ACTH stimulation tests (as mentioned in the discussion section L241). 

Based on the abovementioned data, these may be inferred that serum basal cortisol and the proposed cut-off levels have higher diagnostic performance than serum morning cortisol. 

Point 1 is essential to resolve in my opinion - the cut-off of 420 nmol/L is still at least 20 nmol/L higher than any published data on the cortisol II assay in the ACTH stimulation test, if this was in fact the assay used throughout the study period.

We have addressed and clarified this issue on question#1.

Taken on face value with the current data set, the conclusion of the study should be that a basal cortisol taken any time during the morning up to 1200h (it is unclear how many were done between 1200h and 1300h) provides useful information in assessing the HPA axis integrity, though the cut-offs are not significantly different from the 0800h cortisol.

Thank you for raising this issue, we have discussed this issue in the first paragraph of question#2.

Reviewer #2: My concerns have been addressed. I appreciate the extensive work of the authors in regard to language, statistics and readability.

Thank you. We are really appreciate all your helpful comments which help improved our manuscript.

---

## [Decision Letter · Decision Letter 2]

18 Oct 2019

PONE-D-19-19551R2

Basal cortisol level at 0900 h-1300 h has higher diagnostic performance than the 0800 h morning cortisol level for adrenal insufficiency

PLOS ONE

Dear Dr. manosroi,

Thank you for submitting your manuscript to PLOS ONE. After careful consideration, we feel that it has merit but does not fully meet PLOS ONE’s publication criteria as it currently stands. Therefore, we invite you to submit a revised version of the manuscript that addresses the points raised during the review process.

The reviewers are still not satisfied, especially with your conclusions.

Usually we only allow two revisions, but I find that the findings might have some clinical impact on the more practically level. Accordingly; allow one final revision, but please take the issues raised by the first reviewer into account.

We would appreciate receiving your revised manuscript by Dec 02 2019 11:59PM. To enhance the reproducibility of your results, we recommend that if applicable you deposit your laboratory protocols in protocols.io, where a protocol can be assigned its own identifier (DOI) such that it can be cited independently in the future. For instructions see: http://journals.plos.org/plosone/s/submission-guidelines#loc-laboratory-protocols

We look forward to receiving your revised manuscript.

Kind regards,

Pal Bela Szecsi, M.D. D.M.Sci.

Academic Editor

PLOS ONE

Reviewers' comments:

Reviewer's Responses to Questions

**Comments to the Author**

1. If the authors have adequately addressed your comments raised in a previous round of review and you feel that this manuscript is now acceptable for publication, you may indicate that here to bypass the “Comments to the Author” section, enter your conflict of interest statement in the “Confidential to Editor” section, and submit your "Accept" recommendation.

Reviewer #1: (No Response)

Reviewer #2: All comments have been addressed

2. Is the manuscript technically sound, and do the data support the conclusions?

Reviewer #1: Partly

Reviewer #2: Yes

3. Has the statistical analysis been performed appropriately and rigorously? 

Reviewer #1: Yes

Reviewer #2: Yes

4. Have the authors made all data underlying the findings in their manuscript fully available?

Reviewer #1: Yes

Reviewer #2: Yes

5. Is the manuscript presented in an intelligible fashion and written in standard English?

Reviewer #1: Yes

Reviewer #2: Yes

6. Review Comments to the Author

Reviewer #1: This paper has gone though a number of revisions as the authors sort out exactly how cortisol was measured at their institution. Now they have confined their analysis to the old cortisol I assay and present an entirely new set of results.

1. While I accept the authors' statistical analysis, I do not accept the conclusion that a cortisol done at any time of the morning is inherently superior and a more accurate predictor of adrenal insufficiency than a 0800h cortisol. It makes no sense to me how in the "normal" group, the 0800h cortisol was numerically lower than the basal cortisol (Table 1). I do agree that it is feasible to establish cortisol levels at different times of the day above and below which an ACTH stimulation test is not required to rule in or rule out adrenal insufficiency.

2. As outlined in my previous comments, the proposed cut-offs for "morning" and "basal" cortisol lie well within the overlapping coefficients of variation of the assay. The analysis is based on an assay no longer in use and cannot necessarily be extrapolated to the new cortisol II assay.

3. While acknowledged in the Discussion, mixing the low and standard dose ACTH test and using the same cortisol cut-off for each is a weakness and source of inaccuracy.

4. In the Abstract (Results), please change the cortisol units from mcg/dL to nmol/L - should read 350 nmol/L

5. In Table 1, please remove the decimal points when examining cortisol and other parameters (inappropriate to measure cortisol to 2 decimal points). Furthermore, the way the data are now presented, mean and range would be more illustrative than mean and SD.

Reviewer #2: L.90-91

Those who had serum morning (0800 h) cortisol levels that fell into intermediate levels of

83-499 nmol/L (3-17.915.1 μg/dL) proceeded to either LDT or HDT.

As a retrospective study, and in light that the cut-offs have changed during the review process, it is misleading to state that patients were not subjected to a LDT/HDT. Rather write that these cases were classified as AI/non-AI.

L 246-247 Hence, in terms of using the proposed cut-off levels, the basal cortisol tests appear to be preferable over the 0800 h morning cortisol tests to facilitate AI diagnosis.

I understand my co-reviewers concerns on the question of morning cortisol vs. basal cortisol. There is no considerable difference between these two measurements. Also, clinical application is questionable, as slot for blood draw need to be filled anyway. Therefore, I suggest rephrasing above lines (and similar ones) into a more conciliatory tone. For instance, one could stipulate that the basal cortisol is as good as the morning cortisol, and that there is no specific need for drawing distinct morning cortisol values.

Furthermore, this issue could be alleviated by updating figure 2 with 95% confidence bands around the ROC curves.

Table 1

As your paper is based on the reliability question of basal cortisol, it might be prudent to introduce a row for the time of blood draw in table 1.

7. PLOS authors have the option to publish the peer review history of their article (what does this mean?). If published, this will include your full peer review and any attached files.

Reviewer #1: No

Reviewer #2: Yes: Tristan Struja

---

## [Author Response · Author response to Decision Letter 2]

30 Oct 2019

Response to reviewers

Reviewer #1: This paper has gone though a number of revisions as the authors sort out exactly how cortisol was measured at their institution. Now they have confined their analysis to the old cortisol I assay and present an entirely new set of results.

1. While I accept the authors' statistical analysis, I do not accept the conclusion that a cortisol done at any time of the morning is inherently superior and a more accurate predictor of adrenal insufficiency than a 0800h cortisol. It makes no sense to me how in the "normal" group, the 0800h cortisol was numerically lower than the basal cortisol (Table 1). I do agree that it is feasible to establish cortisol levels at different times of the day above and below which an ACTH stimulation test is not required to rule in or rule out adrenal insufficiency.

We agree with the reviewer that it looks strange that the data in the normal adrenal response group showed that 0800h cortisol was lower than the basal cortisol. We re-checked the raw data and the entire data set again and found that the data shown in Table 1, although perhaps unexpected, is correct. 

Regarding the conclusion, we have modified the sentence in the abstract and have removed comparative words like “superior” and “higher”. We have further noted that basal cortisol can be used as a screening option to diagnose AI as you suggested. Other sentences related to this issue have been similarly modified (L203, 209, 211, 245, 256, 247-248, 259). 

The title has been changed to “Diagnostic Performance of Basal cortisol Level at 0900-1300h in Adrenal Insufficiency” and the word “higher diagnostic performance” has been removed.

2. As outlined in my previous comments, the proposed cut-offs for "morning" and "basal" cortisol lie well within the overlapping coefficients of variation of the assay. 

Thank you for mentioning this. We have modified Table 3 and the interval of the cut-off levels have been changed to more than 10% of the prior cut-off level based on the coefficient of variation of the assay or 25 nmol/L, as appropriate. The statement in the statistical anlysis section has been modified accordingly (L118-L119). 

The analysis is based on an assay no longer in use and cannot necessarily be extrapolated to the new cortisol II assay.

Thank you for your comment. We have included this issue in the Discussion section (limitations). At least, our results can be used as a guide for the future studies using the second generation cortisol assay (L249-L257). 

3. While acknowledged in the Discussion, mixing the low and standard dose ACTH test and using the same cortisol cut-off for each is a weakness and source of inaccuracy.

According to the meta-analysis from Ospina NS, JCEM 2016 (ref. #19), no significant difference in diagnostic accuracy between LDT and HDT has been demonstrated when the same cut-off point is used. However, there are still some debate about this issue. 

In our study, we used ACTH dosage as one of the potentially counfounding factors and adujsted for it using multivariable regression analysis. That adjustment somewhat reduces the confounding potential related to the different ACTH dosages. We have included this potential source of diagnostic inaccuracy as a limitation in the Discussion section (L271-272). 

4. In the Abstract (Results), please change the cortisol units from mcg/dL to nmol/L - should read 350 nmol/L

Thank you. The units have been changed on L32.

5. In Table 1, please remove the decimal points when examining cortisol and other parameters (inappropriate to measure cortisol to 2 decimal points). Furthermore, the way the data are now presented, mean and range would be more illustrative than mean and SD.

Thank you. The data are now shown with one decimal point and are now presented with mean and range as suggested (Table 1). The Statistical Analysis and Results sections have been modified accordingly (L111 and L128)

Reviewer #2: 

L.90-91

Those who had serum morning (0800 h) cortisol levels that fell into intermediate levels of 83-499 nmol/L (3-17.9 μg/dL) proceeded to either LDT or HDT.

As a retrospective study, and in light that the cut-offs have changed during the review process, it is misleading to state that patients were not subjected to a LDT/HDT. Rather write that these cases were classified as AI/non-AI.

Thank you. The sentence has been modified as suggested (L89-L90).

L 246-247 Hence, in terms of using the proposed cut-off levels, the basal cortisol tests appear to be preferable over the 0800 h morning cortisol tests to facilitate AI diagnosis.

I understand my co-reviewers concerns on the question of morning cortisol vs. basal cortisol. There is no considerable difference between these two measurements. Also, clinical application is questionable, as slot for blood draw need to be filled anyway. Therefore, I suggest rephrasing above lines (and similar ones) into a more conciliatory tone. For instance, one could stipulate that the basal cortisol is as good as the morning cortisol, and that there is no specific need for drawing distinct morning cortisol values.

Thank you for the suggestion. We have rephrased the conclusion in the abstract as you suggested and the title of the manuscript has been changed to “Diagnostic Performance of Basal cortisol Level at 0900-1300h in Adrenal Insufficiency”. Other sentences related to this issue have been modified (L203, 209, 211, 245, 255, 247-248, 259). We have used the phrase “statistically higher diagnostic performance,” but in describing the clinical implications we have adopted a more concilialtory tone, stating that basal cortisol level can be employed as a screening option to diagnose AI. 

Furthermore, this issue could be alleviated by updating figure 2 with 95% confidence bands around the ROC curves.

Thank you for the suggestion. The 95%CI bands for ROC curves have been added in Figure 2.

Table 1

As your paper is based on the reliability question of basal cortisol, it might be prudent to introduce a row for the time of blood draw in table 1

Thank you for the suggestion. Unfortunately, as we mentioned in our response to comments on the first version of the manuscript, data regarding the specific time of the test is not available. We have included this issue in the Limitations section (L281-L283).

---

## [Editor Report · Decision Letter 3]

1 Nov 2019

Diagnostic Performance of Basal Cortisol Level at 0900-1300h in Adrenal Insufficiency

PONE-D-19-19551R3

Dear Dr. manosroi,

We are pleased to inform you that your manuscript has been judged scientifically suitable for publication and will be formally accepted for publication once it complies with all outstanding technical requirements.

With kind regards,

Pal Bela Szecsi, M.D. D.M.Sci.

Academic Editor

PLOS ONE
---

## [Editor Report · Acceptance letter]

8 Nov 2019

PONE-D-19-19551R3 

Diagnostic Performance of Basal Cortisol Level at 0900-1300h in Adrenal Insufficiency 

Dear Dr. manosroi:

I am pleased to inform you that your manuscript has been deemed suitable for publication in PLOS ONE. Congratulations! Your manuscript is now with our production department. 

With kind regards,

on behalf of

Dr. Pal Bela Szecsi 

Academic Editor

PLOS ONE